# Dental Neglect and Its Perception in the Dental Practice

**DOI:** 10.3390/ijerph19116408

**Published:** 2022-05-25

**Authors:** Silke Pawils, Tom Lindeman, Rüdiger Lemke

**Affiliations:** 1Center for Psychosocial Medicine, Department of Medical Psychology, University Medical Center Hamburg-Eppendorf, 20246 Hamburg, Germany; lindemanza@gmail.com; 2Center for Dental and Oral Medicine, Department of Periodontics, Preventive and Restorative Dentistry, University Medical Center Hamburg-Eppendorf, 20246 Hamburg, Germany; lemke@uke.de

**Keywords:** dental neglect, detection of child abuse, child protection

## Abstract

In 2020, the number of reported cases of child maltreatment in Germany has never been higher and most of them showed signs of neglect. Most of the time, dental neglect (DN) appears together with a general form of neglect, and includes, apart from caries, many other negative short- and long-term effects for the affected child. In this study, the prevalence of DN in Germany and the way dental practices are currently addressing the topic are examined. Moreover, this study explored whether the experiences of German dentists with DN are related to their work experience, their gender or further education about DN. The data was collected using an anonymous questionnaire. The hypotheses were tested using unpaired *t*-tests based on differences in mean values. The three most common reasons for a suspected case of child maltreatment were the interaction of children with parents, or a legal guardian, insufficient oral or general hygiene, and grave caries. Even though most of the participating dentists agree that it is the task of the dentist to report suspected cases of child maltreatment, only few of them have done so in the past themselves. Not only insecurity about recognition and whom to contact in the suspected case, but also concern about unfounded suspicion were the most common reasons not to report a suspected case of DN. The detection and communication of suspected cases should be encouraged in order to protect affected children.

## 1. Introduction

Neglect, including dental neglect (DN), is a form of child maltreatment and is one of the four forms of child abuse and child endangerment alongside physical abuse, sexual abuse and psychological abuse [1].

The number of reported cases of child maltreatment in Germany reached its peak in 2020, with approximately 60,600 reported cases, which demonstrates the importance of this topic [2].

Unfortunately, there is no specific number of decayed teeth or other oral diseases which inevitably lead to the diagnosis of DN [3]. Moreover, there is no internationally consistent definition of DN [4]. According to the German Child Safeguarding Policy Office, withholding of necessary dental treatment or support of oral hygiene from children are strong indicators for DN [3]. Most of the time, DN appears together with a general form of neglect [5], and includes, besides caries, many other negative short- and long-term effects for the affected child [6]. DN is a neglect of oral health and causes painful inflammation of the oral cavity and decay of the teeth. If left untreated, DN can lead to serious long-term consequences for the child’s health and quality of life, including reduced food intake (which also results in reduced body weight), reduced performance at school, lack of sleep and language problems [7]. These consequences can affect the child’s social life and also their psychological well-being [8]. In addition, children with early childhood caries are more likely to develop caries in their permanent teeth [9].

Nevertheless, dentists must decide whether a child is (dental) neglected during their everyday practice. Other international studies have already shown that many dentists had suspected cases of DN during their careers, but in most cases had not taken any further steps and did not contact other responsible offices [10,11,12,13,14,15]. The main reason was the uncertainty about the diagnosis of suspected cases of child abuse or neglect [7].

In contrast, dentists believe that they are in a good position to identify child abuse or neglect, and that the majority would like to receive further training in this area to successfully identify neglect [16].

This study is intended to give an overview of the perception of dental neglect and how it is being dealt with in dental practices. The aim is to examine in more detail what influences the dental approach to dental neglect such as work experience, further training and gender. In addition, it is analyzed whether participation in training on the subject of DN also leads to less uncertainty in the recognition of DN.

## 2. Materials and Methods

Dentists throughout Germany were invited to answer an online questionnaire that was created using SoSci-Survey. Experiences and handling of dentists with DN and other forms of child maltreatment were quantitatively queried.

### 2.1. Conduct of the Survey

The survey was approved by the psychological Ethics Committee of the university hospital Hamburg-Eppendorf (protocol code: LPEK-0198 and date of approval: 7 October 2020).

Before the actual survey, a pretest was carried out to check the comprehensibility and quality of the questionnaire. The pretest was carried out via SoSci-Survey with five dentists. The suggestions of the participants were implemented.

In 2020, there are 72,500 working dentists in Germany [17]. In order to reach as many dentists as possible, the first step was to contact all state dental associations in Germany, as well as the Federal Dental Association, and ask for their support. For data protection reasons, the chambers could not provide the e-mail addresses of their members for this study which is why the participating chambers published the link to the online survey either in the monthly e-mail newsletter or on their homepage for their members. The survey was online for six weeks (1 September 2020–15 October 2020).

Unfortunately, after this time, the participation rate was insufficient for the planned statistical analyses. For this reason, 5000 randomly selected dentists in Germany (addresses were bought by Scitrace GmbH) were emailed directly on 22 October 2020. About 14 days later, a reminder email was sent again.

### 2.2. Instrument

The dentists contacted were provided with a link to participate. The first pages of the questionnaire explain the relevance of DN for dentists and guarantees an anonymous storage and the data protection concept and the data protection declaration.

If the data protection declaration was agreed, a definition for DN followed, and then the questionnaire with 32 items began.

1. section is about the everyday dealings with DN and questions in which circumstances support and impede the recognition of DN.

2. section is about suspected cases of child maltreatment (physical abuse, sexual abuse, psychological abuse and neglect). With prior permission, the choice of response options was partially taken from the Norwegian questionnaire for the study “Reasons for reported suspicion of child maltreatment and responses from the child welfare—a cross-sectional study of Norwegian public dental health personnel” by Brattabø, Bjørknes, Åstrøm [18].

3. Section is about socio-demographic data of the participants.

In the entire questionnaire, it is possible to answer “don’t know” or “no answer” for each item. Sometimes the participants have the possibility to add free text.

### 2.3. Statistical Method

The information in the questionnaires was transferred to an SPSS table using SoSci (SPSS version 27) (IBM, Armonk, USA). All sample characteristics and results were analyzed and reproduced descriptively, whereby the free text information was previously clustered by the authors and the frequencies of the mentions can be stated afterwards. The comparison of means with regard to further training, professional experience, gender and knowledge of the child protection guideline is carried out using a *t*-test (Fisher). A mean effect is assumed, and the error probability is set at *p* ≤ 0.5. Cohen’s d is calculated to estimate the effect size. In the case of multiple t-testing, an alpha correction is necessary. This is done by performing a Bonferroni correction and lowering the alpha level for each test down to 0.01 since five tests were run.

## 3. Results and Their Interpretation

### 3.1. Description of the Participants

The random sample to be evaluated includes 264 participants (N total). This number of participants reached at least the third page of the questionnaire and thus agreed to the data protection declaration.

As seen in Table 1, 85 of the participants were male, 85 female, and a further 94 were not specified. The average age was 48.5 years (SD = 11.9), and the average previous work experience was 22.0 years (SD = 11.1).

With the exception of Rhineland-Palatinate, the practices of the participants were distributed across all federal states, but very unevenly, e.g., 98 practices were in Lower Saxony alone. This is due to the lack of support for the study from many state dental associations.

The majority of the participants work in an individual practice (56.7%), which is located in a place with 10,000 to 49,000 inhabitants (30.7%) and has a more urban catchment area (38.8%). The vast majority are licensed by health insurance (92.7%) and primarily treat adults (62.7%). Regarding the form of practice, three participants stated in free text that they were employed.

It is important to note that only 10.2% of the participants attended additional qualifications or further training in which the topic of dental neglect was addressed. The majority of them (64.3%) stated that they had learned more about DN in the context of further training.

### 3.2. Knowledge of the Definition of Dental Neglect and Child Endangerment in General

#### 3.2.1. Definition of DN and Knowledge of the Guideline

As seen in Table 2, the definition of DN from the current German child protection guideline (2019) was not known to the vast majority (63.3%). Even fewer knew the part specifically for dentists in this guideline (88.6%). For those familiar with the definition and the dentistry section just mentioned, the guideline helps in identifying DN (60%).

#### 3.2.2. Treated Children and Suspected Cases of Child Endangerment

On average, the participants treated 572 children in 2019 (SD = 1689.9). After the participants were shown the definition of dental neglect, they suspected DN in 37.3 (SD = 99.2) of the children they treated in 2019. In 2019, the participants had an average of three suspected cases of physical abuse (SD = 3.5), 1.9 of sexual abuse (SD = 1.5), 6.9 of psychological abuse (SD = 9.0), and 19.2 suspected cases of general neglect (SD = 50.9).

#### 3.2.3. Reasons for Suspecting Child Endangerment

As seen in Table 3, the three most common reasons for suspecting child endangerment were interaction with parents or guardians, poor oral or general body hygiene, and severe tooth decay.

The main reason given for suspected physical abuse (N = 26) and psychological abuse (N = 49) was the interaction between the child and parents or legal guardians. In the case of general neglect, a large majority chose strong caries as the reason for the suspicion (N = 101). In the case of suspected sexual abuse, “don’t know” (N = 26) and abnormal behavior in the child (N = 12) were chosen most frequently. In the case of free text information, the participants mentioned different characteristics concerning the different forms of child maltreatment:-General neglect: Unkempt appearance and language restrictions (N = 2)-Physical abuse: Hematomas (face or arm) (N = 2)-Sexual abuse: Abnormal feeling of choking when instruments were inserted into the mouth (N = 1)-Psychological abuse: Massive weight loss since the last check-up (N = 1), lack of trust (N = 1)

#### 3.2.4. Referral of Suspected Cases

In Table 4 it can be seen that the majority of participants who suspected one of the forms of child endangerment mentioned above did not refer the suspected case (N = 134, 42%), although the majority of participants see it as the dentist’s task to refer neglected children accordingly (N = 139, 72.4%).

Overall, the most common reasons for not referring were uncertainty about the diagnosis, uncertainty about who to contact in the event of a suspected case, and concern if the suspected case turns out to be unfounded.

In the free text information, the general reason given as a reason not to be referred if there was suspicion of physical or psychological abuse and neglect was that a conversation with the parents was sought instead, or that there was insufficient concern for the well-being of the child. When suspecting general neglect, it was stated, among other things, that there were language barriers, which made communication difficult. One participant also stated that the youth welfare office would not do anything, and that some of the families chosen by the youth welfare offices were even worse, which is why there was no referral. In the free text concerning one case of sexual abuse, it was also stated that the youth welfare office and the police were already aware of the case, which is why it was not passed on.

If the suspected cases were referred, they were usually referred to the youth welfare office, to resident paediatricians, or to counseling centers.

In suspected cases of psychological abuse and general neglect, it was stated in the free text that these cases were referred to specialized pediatric dentists, as well as in suspected cases of general neglect, here above all for oral rehabilitation in general anaesthesia. In cases of psychological abuse, the youth welfare office was also called in several times.

The experience with other actors of state child welfare was mostly described as good (7.8%) and neutral (11.1%); 13.3% rated the cooperation as poorly or very bad. The majority (59.4%) had no experience with state child welfare.

### 3.3. Dental Neglect in Your Practice

#### 3.3.1. Difficulty Dealing with DN

According to the participants, the diagnosis of DN is not made more difficult by an excessive workload or insufficient resources (62%). The majority of participants also state that they have specialist knowledge in the subject area “DN” (48.5%) and are able to deal adequately with the situation (48.5%). In addition, the majority thinks that they are not overwhelmed by being able to diagnose DN (67.2%).

The participants’ assessment of being able to deal with the situation adequately and being able to identify DN with certainty contradicts the result of the previous tables, in which the participants stated that they did not refer a suspected case because they were unsure about the diagnosis. These contradictory results could be related to the fact that people in management positions tend to assess their own competence as above average [19]. Overconfidence is also widespread in medicine [20].

#### 3.3.2. Making It Easier to Deal with DN

Online further training for dentists with CME recognition (45.6%), specialist articles with CME recognition (58.6%), online image material for viewing at the dental associations (64.4%), lectures on the subject of DN already during the study of dentistry (69.1%) and a screening sheet for parents (48.2%) provide support for the majority in recognizing DN. CME is short for Continuing Medical Education, and the point value of the training depends on the evaluation of the federal dental association and the German Society for Dental, Oral and Maxillofacial Medicine [21]. Dentists in Germany are obligated to collect at least 125 CME-points in five years [22].

The positive attitude of the participants towards further training opportunities shows a fundamental interest in further training opportunities on the subject of DN. The results also show that there is currently not enough information on the subject of DN, and possibly also on other forms of child endangerment in Germany. Internationally, too, dentists are generally interested in advanced training opportunities on DN [16].

### 3.4. Analysis of Systematic Connections between Socio-Demographic Variables and the Feedback

In the following, *t*-tests were used to analyze the influence of various sociodemographic factors on the number of suspected cases of DN and also on the uncertainty in detecting DN.

To counteract the alpha error accumulation caused by the multiple tests, a Bonferroni correction was performed. This lowers the alpha level for each test to 0.01 since five tests were run.

#### 3.4.1. Further Training on the Subject of DN and the Influence on the Number of Suspected Cases of DN and on the Uncertainty in the Detection of DN

Participants who attended a training course in which DN was a topic had significantly more suspected cases of DN: they had four times as many (on average 92.8 more) suspected cases in 2019 than the participants who did not attend any training course (95% CI [1.66, 184.03]), t(22.65) = 2.11, *p* ≤ 0.05, Cohen’s d = 0.95).

In addition, the trained participants felt more confident in recognizing DN than participants who had not attended any further training on the subject of DN (95% CI [−0.09, 0.80]) t(171) = 1.59 *p* = 0.11, Cohen’s d= 0.34).

Due to the conservative Bonferroni alpha correction, no significant differences can be assumed between the group that attended further training and the other group that did not attend further training. However, when it comes to interpreting the content, the result indicates that attending further training on the subject of DN promotes the recognition of DN. Cases of dental neglect are often illustrated to the participants of these training courses with important image material, which is very valuable for diagnostics in everyday clinical practice and should also be used in university education.

#### 3.4.2. Differences of Professional Experience and Gender on the Number of Suspected Cases of DN

Years of work experience were dichotomized into a group with fewer or exactly 10 years of work experience and a group with more than 10 years of work experience, so that an analysis for differences in means could be carried out.

The group with less or exactly 10 years of professional experience had an average of 16.9 more suspected cases of DN (95% CI [−19.94, 53.73]) t(154) = 0.91 *p* = 0.37, Cohen’s d = 0.18). There is no statistically significant difference between the two groups with more and less work experience.

Since the early 2000s, many initiatives have been launched in Germany to improve the oral health of children and adolescents [23]. In the course of this development, the oral health of adolescents has also increasingly come into focus in teaching. The generation of dentists with less than 10 years of professional experience has therefore experienced this development during their studies, and is possibly more sensitive to recognizing early childhood caries, which is very frequently associated with DN but is not a primary characteristic according to the definition [7].

On average, women had more than twice as many suspected cases as men (mean 28.23 more cases) (95% CI [−57.99, 1.53]) t(121.19) = −1.88 *p* = 0.06). A statistically significant difference between the two sexes cannot be assumed.

#### 3.4.3. Influence of Knowledge of the Child Protection Guideline on the Uncertainty in the Recognition of DN

Participants who have read the current child protection guideline feel more confident in recognizing DN than participants who have not yet read this guideline (95% CI [−0.02, 1.03]) t(187) = 1.89 *p* = 0.06). In this sample, there was no statistically significant difference between the group that read the guideline and the group that did not read the guideline.

The Child Safeguarding Guideline and the Gown Pocket Card, both published together by the Child Safeguarding Guideline Office, can help identify DN in everyday clinical practice. With regard to the influence of professional experience and gender on the recognition of DN, random differences in the statistical significance test must be assumed. The training shows an influence on the descriptive level, which coincidentally shows a high effect size in the uncorrected significance test for differences in mean values. Participation in advanced training courses on DN seems to increase recognition in professional practice. However, after the conservative alpha correction, the result must be assessed as random, and the hypothesis that experiences of German dentists with DN are related to their professional experience, gender and further training must be rejected.

## 4. Discussion

The current statistics show that dentists are confronted not only with children who are dentally neglected, but also with other forms of child welfare endangerment, in everyday practice. In 2020, the number of child endangerments in Germany reached a new high [2]. In the present study, the participants stated that, in 2019, they suspected dental neglect in an average of 37.3 children and another form of child endangerment in 31 other children. As in the national statistics, in the present study, general neglect, at 61.9%, represents the largest proportion of general forms of child welfare endangerment [2].

The aim of this work was to investigate how German dentists are currently dealing with DN, but also with other forms of child endangerment. For this purpose, a Germany-wide survey among dentists was carried out.

In total, 63.9% of the participants did not know the German definition for DN and 88.6% did not know the part for dentists in the current child protection guideline. This part of the guideline explained both how to recognize DN and who to contact in the case of suspicion, which helped around 60% of the participants who knew the definition and the relevant section to identify DN. Accordingly, they felt more secure than participants who had not read the guideline.

This also clearly explains why the main reason for not referring is uncertainty about the diagnosis (31.3%). Overall, fewer suspected cases were referred, which can be attributed to these uncertainties.

Accordingly, it has already been shown several times in international studies that dentists are reluctant to pass on suspected cases [12,15,24,25]. The main reasons for this are also the uncertainty in diagnostics and the lack of clarity as to who to contact in suspected cases [7,16].

If, in the present study, a referral did take place in the context of a suspected case, the most frequent referral was to the youth welfare office. In order to avert a hazard, according to § 4 of the law on cooperation and information in child protection, members of a healthcare profession have the authority to inform the youth welfare office [3]. If child abuse is suspected, the police also advise informing counseling centers, the youth welfare office and the police themselves [24]. After the youth welfare office, the most frequent referrals were to established pediatricians or counseling centers. The pediatricians have to be specially trained as part of their specialist training in order to be able to recognize endangerments to children’s welfare [25].

The three most common reasons for suspecting child endangerment were interaction with parents or guardians, poor oral or general body hygiene, and severe tooth decay. Severe caries and poor hygiene were also given internationally as the most common reasons for suspecting child endangerment [18]. Parents or legal guardians are responsible for supporting the child’s general hygiene and oral hygiene [26]. If no support is guaranteed, the suspicion of endangering the welfare of the child is obvious. Poor oral hygiene can later lead to severe tooth decay [27]. Mental abuse can result in various long-term consequences, such as loss of trust in adults [28]. This and other forms of child abuse, in turn, have an impact on the interaction between the child and the parents or guardians. For this reason, interaction observation is part of the diagnostics when there is a suspicion of endangering a child’s welfare in the recommendations for action for medical child protection [3].

Although the vast majority is insufficiently informed about DN and feels unsure about diagnosing DN, the majority of participants also state that they have sufficient specialist knowledge in the field of DN and do not feel overwhelmed by being able to diagnose DN with certainty. This contradiction could possibly be related to the fact that people in managerial positions tend to rate their own competence as above average [19]. Overconfidence is also common in medicine [20]. However, this uncertainty about the correct diagnosis and about the correct referral prevents the correct handling of suspected cases of DN, which has already been researched internationally [7].

There are some limitations to the survey results. Self-selection during participation in the survey is likely, as is often the case in voluntary surveys. Dentists who are more interested in the topic and consider themselves competent in the field are more likely to participate in the survey [29].

Because the participants were not randomly written to by the dental associations and not all dental associations took part, the results are also less representative. It is also possible that there are still dentists in Germany who do not have an email address. These should also have been included in order to increase the representativeness of the study. Although an additional 5000 randomly selected dentists were subsequently written to, the overall response rate was very low. This also shows how little interest there is in DN in Germany.

## 5. Conclusions

The results have shown that there is a need for action in Germany, as well as internationally, to help dentists to identify and forward cases of child endangerment such as DN. The majority of the participants had a positive attitude towards the proposed options and acknowledge the responsibility of the dentist to recognize and report suspected cases of child endangerment. It has also been shown that dentists who have attended advanced training courses on the subject of DN felt more confident in recognizing DN and also had more suspected cases.

However, the dentists should know the various signs of DN and of other forms of child endangerment, and then know how to proceed in the event of a justified suspicion. If this does not endanger the protection of the child, this includes talking to the parents or legal guardians and the child and, if necessary, discussing the case with a paediatrician and/or the medical child protection hotline.

Introducing a screening questionnaire for the parent or guardian and the child would also be a good way to identify risk factors for possible DN or other forms of child endangerment. This was also confirmed by the majority of the participants.

Finally, the main recommendation is to introduce a general reporting obligation for all members of a healthcare profession if there is a strong suspicion of a child endangerment, as is already the case in Norway, since the legal obligation leads to increased mediation of suspected cases [30].

## Figures and Tables

**Table 1 ijerph-19-06408-t001:** Sociodemographics.

Content	Item	Quantity	Frequency
Gender	Male	85	48.0
	Feminine	85	48.0
	Missing	94	35.6
Federal state	Baden-Wuerttemberg	11	6.2
	Bavaria	11	6.2
	Berlin	8	4.5
	Brandenburg	4	2.3
	Bremen	1	0.6
	Hamburg	11	6.2
	Hesse	9	5.1
	Mecklenburg-Western Pomerania	2	1.1
	Lower Saxony	98	55.4
	North Rhine-Westphalia	9	5.1
	Saarland	1	0.6
	Saxony	5	2.8
	Saxony-Anhalt	2	1.1
	Schleswig Holstein	1	0.6
	Thuringia	4	2.3
	Missing	87	33.0
Practice type	Individual practice	101	56.7
	Community of practice	19	10.7
	Professional community/group practice	45	25.3
	Medical care center	4	2.2
	Hospital/clinic	5	2.8
	Miscellaneous	4	2.2
	Missing	86	32.6
Residents place practice	Under 10,000	39	22.2
	10,000–49,000	54	30.7
	50,000–250,000	39	22.2
	More than 250,000	44	25.0
	Not specified	3	1.7
	Missing	88	33.3
Catchment area	Rather urban	69	38.8
	Rather rural	57	32.0
	Both	52	29.2
	Missing	86	32.6
Licensed by health insurance	Yes	165	92.7
	no	10	5.6
	Not specified	3	1.7
	Missing	86	32.6
Treatment focus	Children/teenagers (up to 18 years)	20	11.3
	Adult	111	62.7
	About the same number of adults as children	43	24.3
	Not specified	3	1.7
	Missing	87	33.0
Additional qualifications/ advanced training	Pediatric Dentistry (M.Sc.)	1	3.6
	Curriculum in pediatric and adolescent dentistry	8	28.6
	Further training	18	64.3
	Not specified	1	3.6
	Missing	238	90.1
Age (in years)	M(SD) in years	48.5 (11.9)	
Professional activity (in years)	M(SD) in years	22.0 (11.1)	

**Table 2 ijerph-19-06408-t002:** Definition of dental neglect (DN) and knowledge of the child protection guideline (N total = 264).

Content	Item	Quantity	Frequency
Knowledge of the definition	Yes	66	28.8
	No	145	63.3
	I do not know	18	7.8
	Missing	35	
Knowledge of the section for dentists in the child protection guideline	Yes	22	10.0
	No	195	88.6
	I do not know	3	1.4
	Missing	44	
Guideline helpful in recognizing DN	Yes	12	60.0
	No	5	25.0
	I do not know	3	15.0
	Missing	244	

**Table 3 ijerph-19-06408-t003:** Reasons for suspected child maltreatment (N total = 218).

Content	Total		Physical Abuse		Sexual Abuse		Mental Abuse		Neglect	
	N = 218	%	N = 72	%	N = 54	%	N = 80	%	N = 128	%
Interaction with parents/guardians	156	71.6	26	36.1	11	20.4	49	61.3	70	54.7
Poor (oral) hygiene	142	65.1	19	26.4	3	5.6	22	27.5	98	76.6
Severe tooth decay	136	62.4	17	23.6	0	0	18	22.5	101	78.9
Repeatedly missed appointments	131	60.1	14	19.4	3	5.6	22	27.5	92	71.9
Abnormal behavior in the child	127	58.3	22	30.6	12	22.2	40	50.0	53	41.4
Treatment refusal	106	48.6	15	20.8	8	14.8	34	42.3	49	38.3
Gingivitis	78	35.8	11	15.3	1	1.9	10	12.5	56	43.8
Miscellaneous	49	22.5	13	18.1	12	22.2	10	12.5	14	10.9
Trauma	32	14.7	16	22.2	3	5.6	3	3.8	10	7.8
Ulcers and lesions in the oral cavity	15	6.9	1	1.4	0	0	3	3.8	11	8.6
Other oral findings	15	6.9	1	1.4	2	3.7	2	2.5	10	7.8
I do not know	64	29.4	18	25.0	26	48.1	16	20	4	3.1
Missing			192		210		184		136	

**Table 4 ijerph-19-06408-t004:** Referral of suspected cases (N total = 319).

Content	Total	Physical Abuse	Sexual Abuse	Mental Abuse	Neglect
N = 319	%	N = 68	%	N = 51	%	N = 76	%	N = 124	%
Transferred to …	92	28.8	17	25.0	8	15.7	21	27.6	46	37.1
Youth welfare office	37	40.2	10	58.8	6	75.0	11	52.4	10	21.7
Resident paediatrician	35	38.0	7	41.2	2	25.0	6	28.6	20	43.5
Counseling centers	20	21.7	3	17.7	3	37.5	6	28.6	8	17.4
Miscellaneous	15	16.3	2	11.8	1	12.5	3	14.3	9	19.6
Specialized ambulances (e.g., trauma ambulance)	8	8.7	1	5.9	3	37.5	2	9.5	2	4.4
Police	7	7.6	2	11.8	3	37.5	1	4.8	1	2.2
Established (child and youth) psychotherapists	6	6.5	1	5.9	2	25.0	2	9.5	1	2.2
Established specialist for child and adolescent psychiatry and psychotherapy	5	5.4	0	0	1	12.5	2	9.5	2	4.4
Inpatient facilities for child and adolescent psychiatry and psychotherapy	4	4.4	1	5.9	1	12.5	1	4.8	1	2.2
Not transferred because …	134	42.0	24	35.3	12	23.5	32	42.1	66	53.2
Unsure about diagnosis	42	31.3	8	33.3	4	33.3	9	28.1	21	31.8
Unsure who to contact about suspicion	42	31.3	6	25	2	16.7	11	34.4	23	34.9
Worried if suspicions prove unfounded	38	28.4	8	33.3	2	16.7	11	34.4	17	25.8
Unsure how to report suspicion	38	28.4	7	29.2	2	16.7	10	31.3	19	28.8
Youth Welfare already involved	34	25.4	5	20.8	3	25	11	34.4	15	22.7
Not enough knowledge about child abuse	31	23.1	2	8.3	2	16.7	7	21.9	20	30.3
I don’t feel trained for this	27	20.2	4	16.7	0	0	9	28.1	14	21.2
No routine when reporting/forwarding	26	19.4	4	16.7	2	16.7	6	18.8	14	21.2
Unsure of the implications for family	25	18.7	4	16.7	1	8.3	7	21.9	13	19.7
Consider how the child’s parents will react	25	18.7	6	25	1	8.3	5	15.6	13	19.7
Ensure that reporting does not remain anonymous	25	18.7	5	20.8	2	16.7	9	28.1	9	13.6
Hampered by confidentiality	24	17.9	5	20.8	1	8.3	6	18.8	12	18.2
Miscellaneous	21	15.7	3	12.5	4	33.3	6	18.8	8	12.1
Worry about damage to reputation	16	11.9	3	12.5	1	8.3	5	15.6	7	10.6
Couldn’t discuss suspicions with anyone	14	10.5	1	4.2	2	16.7	3	9.4	8	12.1
Make sure that the child no longer comes to my practice	14	10.5	2	8.3	1	8.3	5	15.6	6	9.1
Worry about being threatened	9	6.7	2	8.3	1	8.3	3	9.4	3	4.6
No support from supervisors	6	4.5	1	4.2	1	8.3	2	6.3	2	3.0
Not my responsibility as a dentist	4	3.0	0	0	1	8.3	1	3.1	2	3.0
I do not know	68	21.3	19	27.9	22	43.1	16	21.1	11	8.9
Missing			196		213		188		140

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
