# Peer review of "Dental Neglect and Its Perception in the Dental Practice"

_ijerph, 2022, doi:10.3390/ijerph19116408_

Round 1

Reviewer 1 Report

Dear Authors, 

this paper is overall well written and interesting.

Some issues are present.

Abstract: in the abstract section, no references should be present, please delete them and start numbering references from introduction.

Introduction: introduction is well written but it is really short, you should add some information regarding also how information and neglect can affect dental health and traumatology to complete introduction; you can use this reference: Ludovichetti FS, Signoriello AG, Zuccon A, Padovani S, Mazzoleni S. The Role of Information in Dental Traumatology in Patients during Developmental Age: A Cognitive Investigation. Eur J Dent. 2021 Oct 22. 

Materials and methods: really well written, easy to understand, very clear

Results: Well presented, interesting and clear

Discussion and conclusion: Discussion in well written but conclusion is really too long. Conclusion of the study should just state some bullet point were you briefly describe the main findings of the study. please, shorten them

Thank you

Reviewer 2 Report

  • ethical approval for conducting research: is not mentioned;
  • references: 15 of 29 references are older than 5 years;
  • I consider that the text from conclusion, between the lines 345-354, are better to be moved  to the discussion section.

Reviewer 3 Report

Dear Author, 

Thank you very much for your interesting work. This is very important topic, and should be covered properly. Therefore, there are some points where you can improve the content of your manuscript.

Firstly, as a reader I do not have any clue what is the total number of dentist in Germany, and what your number of participants is out of that number of dentists in percentages. This is very important to know what is your sample size, measured according to what... I really do not know whether 264 dentists in total is small sample size, is it 10% of total number...

I do not know if this complicated gender representation (missing, miscellaneous, not specified) is of any importance in conclusion. You stated just female versus male response. I think you need to correct this.

And my biggest concern is why you recruited dentist in Germany randomly. Why you did not select dentist working with the kids (pedodontics or general dentists) only? I do not know why you selected dentists working with adults only to participate in this study at all. This is very important topic. There are few published papers related to this topic, therefore initial papers dealing with DN, such as yours, should be carefully examined. The results should be solid in order to recommend guidelines for dentists in case that they will detect DN. Therefore, this study should examine the number of DN in 2019 of people who are actually working with kids.

In results you said 572, 3 children were treated in 2019. But, maybe just a few dentists from your study perform it. I do not think it is stated clearly. Therefore, I think it is important to include dentist working with kids in their everyday practice only

Reviewer 4 Report

Please, see attached file

Round 2

Reviewer 4 Report

I believe that the Authors have done a good job following the suggested revisions and that the paper in the current form is suitable for publication.